# Quantitative Proteogenomic Characterization of Inflamed Murine Colon Tissue Using an Integrated Discovery, Verification, and Validation Proteogenomic Workflow

**DOI:** 10.3390/proteomes10020011

**Published:** 2022-04-14

**Authors:** Andrew T. Rajczewski, Qiyuan Han, Subina Mehta, Praveen Kumar, Pratik D. Jagtap, Charles G. Knutson, James G. Fox, Natalia Y. Tretyakova, Timothy J. Griffin

**Affiliations:** 1Department of Biochemistry, Molecular Biology and Biophysics, University of Minnesota, Minneapolis, MN 55455, USA; rajcz001@umn.edu (A.T.R.); hanxx963@umn.edu (Q.H.); smehta@umn.edu (S.M.); pravs3683@gmail.com (P.K.); pjagtap@umn.edu (P.D.J.); 2Department of Biological Engineering, Massachusetts Institute of Technology, Cambridge, MA 02139, USA; charlie.knutson@novartis.com (C.G.K.); jgfox@mit.edu (J.G.F.); 3Department of Medicinal Chemistry, the Masonic Cancer Center, University of Minnesota, Minneapolis, MN 55455, USA; trety001@umn.edu

**Keywords:** inflammation, proteogenomics, bioinformatics, colon cancer

## Abstract

Chronic inflammation of the colon causes genomic and/or transcriptomic events, which can lead to expression of non-canonical protein sequences contributing to oncogenesis. To better understand these mechanisms, *Rag2*^−/−^*Il10*^−/−^ mice were infected with *Helicobacter hepaticus* to induce chronic inflammation of the cecum and the colon. Transcriptomic data from harvested proximal colon samples were used to generate a customized FASTA database containing non-canonical protein sequences. Using a proteogenomic approach, mass spectrometry data for proximal colon proteins were searched against this custom FASTA database using the Galaxy for Proteomics (Galaxy-P) platform. In addition to the increased abundance in inflammatory response proteins, we also discovered several non-canonical peptide sequences derived from unique proteoforms. We confirmed the veracity of these novel sequences using an automated bioinformatics verification workflow with targeted MS-based assays for peptide validation. Our bioinformatics discovery workflow identified 235 putative non-canonical peptide sequences, of which 58 were verified with high confidence and 39 were validated in targeted proteomics assays. This study provides insights into challenges faced when identifying non-canonical peptides using a proteogenomics approach and demonstrates an integrated workflow addressing these challenges. Our bioinformatic discovery and verification workflow is publicly available and accessible via the Galaxy platform and should be valuable in non-canonical peptide identification using proteogenomics.

## 1. Introduction

Chronic inflammation has been linked to the development of many serious health problems, notably oncogenesis in several tissue types including those related to colorectal cancer [1,2]. During inflammation, continued release of regulatory cytokines which serve to mediate the immune response promotes tumorigenesis [3] and eventual metastasis [4]. In addition, chronic inflammation causes a burst of reactive oxygen species (ROS) and reactive nitrogen species (RNS) which can damage the host genome, contributing to oncogenesis via DNA damage and mutagenesis [5,6]. The full picture of molecular changes which occur during chronic colon inflammation is of interest to advance our understanding of colorectal cancer etiology [1], as well as to seek opportunities for its diagnosis [7] and identification of therapeutic targets for its treatment [8].

Modern omics technologies such as next-generation RNA sequencing (RNA-Seq) and mass spectrometry (MS)-based proteomics have allowed for marked advancements in studies of cancer [9,10]. However, RNA-Seq is only able to assess the state of the transcriptome, which often does not match the expressed proteins (the proteome) associated with a specific tissue or disease state [11]. By contrast, MS-based proteomics can be used to quantitatively assess protein abundance in tumors relative to healthy tissue, as well as to identify cancer biomarkers for early diagnosis and treatment [12].

In conventional “bottom-up” proteomics, MS data are searched against a reference FASTA database containing protein sequences encoded in canonical gene sequences for the organism of interest, thereby excluding any proteins containing non-canonical sequences stemming from insertions, deletions, amino-acid substitutions, alternate splicing events, or any other atypical events leading to translation of proteins with unexpected amino acid sequences [13]. These non-canonical sequences are captured in RNA-Seq analyses, which detect and sequence all transcripts including those that may give rise to novel protein products.

Proteogenomics is a multi-omics approach which combines the comprehensive nature of RNA-Seq with the ability of MS-based proteomics to directly confirm the translation of these products into expressed proteins with potential functional implications, creating a more complete molecular picture of phenotypes as compared with a single omics technology [14,15]. Proteogenomics uses RNA-Seq data to generate an expanded protein sequence FASTA database, which can be used to confirm the expression of proteoforms [16] containing both canonical and novel non-canonical peptide sequences. Although proteogenomics has been shown to be a powerful approach for studying cancer [15,17], potential false-positive matches to non-canonical sequences remains a concern [18], requiring methods to verify the accuracy of PSMs using bioinformatic and/or analytic approaches. To aid in analysis, these assorted bioinformatics processes can be combined into simple workflows for automated, streamlined proteogenomic analyses [19].

In this study, we developed and utilized novel proteogenomic workflows to analyze chronic inflammation in proximal colon tissues in a mouse model of inflammatory bowel disease (IBD). Genetically engineered *Rag2*^−/−^*Il10*^−/−^ mice have been used in previous studies as models of chronic inflammation [20,21], as animals with these mutations develop chronic colon inflammation when subjected to bacterial infection [22]. *Rag2*^−/−^*Il10*^−/−^ mice were subjected to infection with *Helicobacter hepaticus* and allowed to develop chronic colon inflammation as described previously in Mangerich et al. [5], after which proximal colon tissues were harvested and proteins were isolated based off previously established protocols [23] for LC–MS analysis. Using the Galaxy for Proteomics (Galaxy-P) software suite, [24] we utilized two automated computational workflows to generate and refine [25] a transcriptome-derived FASTA database for proteogenomic analysis of the MS data. Finally, a rigorous bioinformatic workflow coupled with targeted MS methods was used to verify and validate non-canonical peptides. In total, our results provide unique insights into molecular signatures of inflammation in the colon and demonstrate a powerful proteogenomic pipeline for verification and validation of novel, non-canonical sequences derived from proteoforms underlying cancer-driving disease phenotypes.

## 2. Materials and Methods

### 2.1. Materials

Proximal colon tissues were obtained from a previous study [5]. Triethylammonium bicarbonate (TEAB), urea, aprotinin, phenylmethanesulfonylfluoride (PMSF), 4-(2-hydroxyethyl)-1-piperazineethanesulfonic acid (HEPES), dithiothreitol (DTT) and iodoacetamide (IAA) were obtained from Millipore Sigma (Burlington, MA, USA). Trypsin was purchased from Promega Corporation (Madison, WI, USA). Formic acid was purchased from Honeywell Fluka (Mexico City, Mexico). Acetonitrile, water, and LTQ ESI Positive Ion Calibration Solution were obtained from Thermo Fisher Scientific (Waltham, MA, USA). Anhydrous acetonitrile was obtained from Glen Research (Sterling, VA, USA).

Kimble 1.5 mL pestles were purchased from VWR International (Radnor, PA, USA). Pall 10 K Nanosep spin filters were utilized for digestion and were obtained from Millipore Sigma (Burlington, MA, USA). Pierce BCA Assay and Colorimetric Peptide Assay kits were obtained from Thermo Fisher Scientific (Waltham, MA, USA). For isobaric labeling, a TMTsixplex kit (lot #SH253249) was purchased from Thermo Fisher Scientific (Waltham, MA, USA). For peptide desalting and fractionation, the Pierce High pH Fractionation kit was obtained from Thermo Fisher Scientific (Waltham, MA, USA).

### 2.2. Treatment Conditions, Tissue and Protein Isolation and Proteolytic Digestion

*Rag2*^−/−^*Il10*^−/−^ mice were subjected to three oral gavages over the course of one week with either saline (control) or *Helicobacter hepaticus* culture, after which the infected mice developed chronic colorectal inflammation (Figure 1) [5].

After 20 weeks, the mice were sacrificed, and colon tissues were collected for subsequent analysis. Our experiments utilized approximately 10 mg of proximal colon tissue harvested from control and infected mice (Appendix A). These samples were placed in individual Eppendorf tubes containing 100 µL of lysis buffer (25 mM TEAB, 8 M urea, 1 mM PMSF, and 2.5 μg/mL aprotinin, pH = 8.5) and disrupted via grinding using 1.5 mL Kettle pestles. After homogenization, samples were subjected to probe sonication at 30% amplitude for 10 s over ice to lyse the cells; following lysis, samples were centrifuged at 15,000 rpm at 4 °C for 15 min, after which the protein content was measured via Pierce BCA Assay. From each sample, 100 μg aliquots of protein were added to individual Pall Nanosep 10 K spin columns. The lysis buffer was then removed via centrifugation at 14,000× *g* for 5 min, followed by the addition of 100 μL of dilution buffer (25 mM TEAB, pH = 8.5). This was repeated twice more to remove the lysis buffer, with the proteins finally reconstituted in 100 µL of dilution buffer. The proteins were then reduced via the addition of 20 μL of DTT in the dilution buffer, followed by incubation at 55 °C for one hour. Samples were then alkylated with the addition of 10 μL of 375 mM IAA to the spin columns, followed by a 30-min incubation in the dark at room temperature. After alkylation, samples were then washed with a further three iterations of centrifugation and the addition of 100 µL of dilution buffer. Samples were finally reconstituted with 50 µL of dilution buffer, to which 4 µg of trypsin was added, and incubated at 37 °C overnight. Following incubation, peptide samples were isolated by spinning the samples through the column filters. A further 50 µL of digestion buffer was then added to the top of the spin columns and spun through via centrifugation. The peptide solution was then transferred to a fresh tube and the concentration determined through a peptide colorimetric assay; 10 µg of peptides from each sample were then aliquoted into fresh vials and dried overnight under vacuum.

### 2.3. Peptide Labeling, Fractionation, and LC–MS/MS Analysis

Peptides were labeled with TMT six-plex reagents for quantitative analysis. One dried-down aliquot of 10 µg from each sample was selected and reconstituted in 35 µL of 100 mM HEPES, pH = 8.0. At the same time, TMT six-plex vials were brought to room temperature, after which the individual labels were reconstituted in 41 µL of anhydrous acetonitrile. Each peptide sample was then labeled via the addition of 10 µL of TMT labeling reagent (Appendix A). The samples were then allowed to incubate for 2 h at room temperature, after which the reaction was terminated via the addition of 4 µL of 5% hydroxylamine and a further 15 min incubation.

Following incubation, the peptide concentrations of each labeled sample were measured; thereafter, 5 µg of each of the six digested samples were concatenated into a single sample containing an equal amount of each of the labeled control and inflamed samples. The pooled sample was then desalted and fractionated using the Pierce High pH Fractionation Spin Columns using mobile phases containing 0.1% triethylamine and increasing amounts of acetonitrile into eight fractions. For each of six samples, eight HPLC fractions were collected, dried down under reduced vacuum, and reconstituted in 10 µL water containing 0.1% formic acid.

The eight fractionated peptide samples were analyzed on an Orbitrap Fusion Tribrid Mass Spectrometer interfaced with an Ultimate 3000 UHPLC. The Fusion LC–MS was calibrated in positive mode using LTQ ESI Positive Ion Calibration Solution. The UHPLC was run in nanoflow mode with a reverse-phase nanoLC column (35 cm × 250 μm) packed with 5 μm diameter Luna C18 resin. Samples were run on a 90-min gradient with 5–22% buffer B (0.1% FA in acetonitrile) over 71 min, followed by 22–33% over 5 min, 33–90% over 5 min, a 90% buffer B wash for 4 min, and finally a 90–4% decrease in buffer B over 2 min followed by a 3-min equilibration at 4% buffer B. Samples were run at a flow rate of 300 nL/min. Peptides were analyzed in positive mode using a Top12 Full MS/dd-MS2 experiment with an expected chromatographic peak FWHM of 15 s. In the full scan mode, resolution was 70,000 with an AGC target of 1 × 10^6^, a maximum IT of 30 ms, and a scan range of 300 to 2000 *m*/*z*. Tandem mass spectrometry experiments were conducted at 17,500 resolution, AGC target of 5 × 10^4^, maximum IT of 50 ms, an isolation window of 2.0 *m*/*z*, an exclusion time of 30 s, and a normalized collision energy of 30. Data were collected in the centroid mode.

### 2.4. Database Construction

Computational work was performed using proteogenomics workflows and tools in the Galaxy for Proteomics (Galaxy-P) suite [26,27] as well as in Proteome Discoverer v2.2 (Thermo Fisher Scientific (Waltham, MA, USA)).

Raw RNA sequencing data were acquired from proximal colon samples of six additional mice from the colon inflammation study (QIYUAN), including three control and three inflamed samples (Figure 1a).

Sequencing data were collected at the University of Minnesota Genomics Center on an Illumina HiSeq 2500 (Illumina, San Diego, CA, USA) sequencer run in high output mode using 50 bp paired end reads. These data were uploaded into Galaxy-P and used as an input for an integrated workflow [26] to generate a customized proteogenomic FASTA database. Briefly, the FASTQ files generated from these samples were paired with a murine genome annotation file and aligned via HISAT2 [28] (v2.1.0, Kim lab, UT Southwestern, Dallas, TX, USA); this was then used to create a list of genetic variants using the Free Bayes (v1.1.0.46-0, Garrison lab, University of Tennessee Health Science Center, Memphis, TN, USA) Bayesian genetic variant detector [29]. This file was then utilized by the CustomProDB (v1.16.1.0, Zhang lab, Baylor College of Medicine, Houston, TX, USA) tool [30] to create FASTA sequences of the mapped indel, single amino acid variants, and alternatively spliced sequences identified. These variants were then concatenated together with the canonical murine Uniprot FASTA database and a list of common mass spectrometry contaminants [31] as a custom RNA-Seq-based database. This workflow also used StringTie [32] (v1.3.3.1, Center for Computational Biology at Johns Hopkins University, Baltimore, MD, USA) to create an assembled gene transfer format file which was used to create a set of genomic coordinates complementary to the RNA-Seq FASTA database used in downstream applications [33], and effective for annotating other types of non-canonical transcripts not handled by CustomProDB.

### 2.5. Database Sectioning

The custom protein FASTA database was matched to MS/MS data to generate PSMs using a sectioning workflow created by Kumar et al. [25] (Figure 1b), which provides increased sensitivity when working with large sequence databases, while controlling false positives. The protein sequences in the database were randomly sorted into five smaller sections; each of these was used to search against the raw mass spectrometry data of the proximal colon samples using Search GUI [34] (v3.3.3.0, CompOmics, VIB-UGent Center for Medical Biotechnology at Ghent University, Ghent, Belgium), with N-terminal and lysine TMT-6 labeling, as well as cysteine carbamidomethylation being set as static modifications, while methionine oxidation and phosphorylation at serine, threonine, and tyrosine were set as dynamic modifications. The X! Tandem search engine was used to identify peptides from the data against the individual batches. These results were then used by PeptideShaker [35] (v1.16.4, CompOmics, VIB-UGent Center for Medical Biotechnology at Ghent University, Ghent, Belgium) to identify proteins in the data against the individual batches. With the resulting PSM report, the proteins in each batch that were identified in the raw data with any level of confidence were retained, while the rest were discarded. For each protein in the batch that was retained, a discarded sequence was then selected at random and added back to the sectioned database. The five sections were then recombined back together to create a compact custom FASTA database enriched for protein sequences found in the inflamed colon samples, which were then in turn concatenated together with the murine UniprotKB and contaminant sequence FASTA databases with redundant proteins removed.

### 2.6. Differential Abundance Proteomic and Proteogenomic Analysis

Raw mass spectrometry files were analyzed using Proteome Discoverer v2.2 (Thermo Fisher Scientific, Waltham, MA, USA) in the TMT6 quantitation mode. The eight raw files were processed utilizing the basic Proteome Discoverer processing and consensus workflows designed for reporter ion quantitation. The murine SwissProt FASTA database was utilized for proteomics analysis, while the sectioned custom FASTA database with the RNA-Seq data-derived sequences was used for proteogenomics analysis. In all instances, carbamidomethylation at cysteine and TMT6 labeling at peptide N-termini and lysine residues were set as static modifications, while methionine oxidation and phosphorylation at serine, threonine, and tyrosine were set as dynamic modifications. Confidence for peptide identifications was set at an FDR cutoff of 0.01. The resulting PSM reports were used for quantitative analysis using MSstatsTMT (Vitek lab, Northeastern University, Boston, MA, USA) [36] using the “mstats” normalization algorithm. Gene ontology analyses were performed using the g-profiler package (Vilo lab, University of Tartu, Tartu, Estonia) [37], using an FDR cutoff of 0.05.

### 2.7. Identification, Verification and Validation of Non-Canonical Peptides

Given the large numbers of proteins, the annotation of non-canonical peptides is more efficiently performed using an automated workflow in the Galaxy-P platform (Figure 1c). As with the sectioning workflow, the raw mass spectrometry data of the proximal colon tissue were searched against the custom protein FASTA database using SearchGUI and PeptideShaker. From the peptides that were identified, peptides from the murine reference and common contaminant reference proteomes were removed, leaving only potential non-canonical peptide sequences resulting from translation of unexpected genomic regions, novel splicing events or amino acid coding sequence variants. These were then searched against the NCBI mouse proteome using Basic Local Alignment Search for Proteins (BLAST-P) [38]; these results were filtered to look for those search results which had imperfect sequence alignments due to sequence substitutions or gaps in the sequence [27]. The genomic coordinates of these peptides were then determined using the PepPointer tool [39] for further analysis and interrogation. Upon completion of the workflow, the identified non-canonical peptides were processed through an automated computational verification step using the PepQuery [40] tool with unrestricted modification search mode and amino acid substitution mode engaged. Peptides were deemed to be valid if they had no matches to reference mouse or random peptides, had a *p*-value < 0.05, and no better scoring matches to any other peptides, such as reference peptides carrying a PTM. For PepQuery analysis, carbamidomethylation of cysteine residues as well as TMT-6 labeling of N-termini and lysine residues were all set as fixed modifications, while phosphorylation of serine, threonine, and tyrosine residues were set as variable residues.

### 2.8. Validation and Quantitation of Non-Canonical Peptides

Peptides verified using PepQuery were further validated by targeted mass spectrometry analyses [41] using 10 μg aliquots of unlabeled peptides reserved from the initial sample processing. The *m*/*z* values for molecular ions and MS/MS product ions of non-canonical peptides were determined from the original global analysis data and used to populate an inclusion list for use in targeted analyses (Appendix A). For targeted analysis, samples were run on a Q-Exactive Hybrid Quadrupole–Orbitrap Mass Spectrometer interfaced with an Ultimate 3000 UHPLC run in nanoflow mode equipped with a nanocolumn packed with 5 μm diameter Luna C18 resin (15 cm × 250 μm). The Q-Exactive was calibrated in positive mode using LTQ ESI Positive Ion Calibration Solution. Samples were run on a 90-min gradient with 5–22% buffer B (0.1% FA in acetonitrile) over 71 min, followed by 22–33% over 5 min, 33–90% over 5 min, a 90% buffer B wash for 4 min, and finally a 90–4% decrease in buffer B over 2 min, followed by a 3-min equilibration at 4% buffer B. HPLC was conducted at a flow rate of 300 nL/min. The mass spectrometer was run in dual Full Scan and Parallel Reaction Monitoring mode. In the full MS, resolution was 70,000 with an AGC target of 3 × 10^6^, a maximum IT of 200 ms, and a scan range of 400 to 1600 *m*/*z*. Parallel reaction monitoring experiments were conducted at a 35,000 resolution, an AGC target of 2 × 10^5^, maximum IT of 100 ms, an isolation window of 4.0 *m*/*z*, an exclusion time of 30 s, and a normalized collision energy of 35. The resulting spectra were then analyzed in Skyline [42] against a spectral library of non-canonical peptides generated using Prosit [43]. Non-canonical peptides were identified by Skyline with at least three b- and/or y-ions, with peak areas of the detected product ions summed to represent the abundance of the peptide. The non-canonical peptide abundances were then tested for differential abundance using limma in R.

For comparison of differential abundance levels of non-canonical peptides with their complementary mRNA levels, the original RNA-Seq data were run through a workflow in the Galaxy-P platform to perform differential transcriptomic analysis. Briefly, paired-end raw FASTQ files were cleaned up using Trimmomatic [44] to remove sequencing adaptors and aligned to the GRCm38 mm10 genome using HiSat2; the resulting BAM files were then assembled and quantified using Stringtie. The resulting transcript counts were then subjected to differential analysis using edgeR [45].

## 3. Results

### 3.1. Creation and Sectioning of a Custom RNA-Seq-Based FASTA Database

Six sets of paired-end RNA-Seq data were obtained by sequencing RNA isolated from the proximal colons of *Rag2*^−/−^*Il10*^−/−^ mice subjected to five months of *H. hepaticus*-induced inflammation along with matching controls (three animals per group, see Figure 1) [21]. Each of these datasets was aligned and mapped to the mm10 mouse genome to create transcriptomic data for these samples; these individual sets of transcriptomic data were then converted to FASTA files representing the proteins that could potentially be translated from the sequencing data (Figure 1a). Concatenating these data together gave a combined RNA Seq-derived database that contained 1,402,947 sequences, corresponding to 1,348,407 protein sequences beyond the canonical mouse FASTA database.

As the large size of the RNA Seq-derived FASTA database increased the likelihood of false positive PSMs while decreasing overall sensitivity for true positive PSMs [46], a sectioning workflow [25] was utilized to create a reduced RNA-Seq-based FASTA database (Figure 1b). Use of the sectioning workflow reduced the RNA-Seq-derived FASTA database down to 423,071 protein sequences. Given that the workflow combines novel protein sequences detected in the raw data with an equivalent number of random sequences, the sectioned database corresponds to approximately 184,266 proteins containing non-canonical portions of their sequences derived from RNA sequences having PSMS in the proteomics data.

### 3.2. Global Proteogenomic Analysis Reveals Inflammation-Driven Changes in Protein Abundance

The reduced, sectioned proteogenomic FASTA database was merged with the reference mouse Uniprot database and the database of common MS contaminants, and the resulting merged database (proteogenomic database) was uploaded into Proteome Discoverer for global quantitative proteomic analysis of the inflamed proximal colon samples. For comparison, the mouse SwissProt FASTA database supplemented with common protein contaminants was also searched against the MS/MS data, offering a more conventional proteomic approach using a reference sequence database. Analysis of TMT-labeled peptides using the proteogenomic database identified 16,725 proteins in the proximal colon data grouped into 4865 protein groups. Of these protein groups, most were annotated proteins corresponding to entries within the mouse SwissProt FASTA database (91.7%). The rest of the identifications corresponded primarily to proteins containing non-canonical sequences generated in the database creation workflow in the Galaxy-P platform, with at least one peptide sequence identified as a part of the protein having a non-canonical sequence. Five of these identified protein groups corresponded to annotated proteins containing non-canonical sequences such as amino acid substitutions; 386 identified protein groups correspond to potentially novel proteins annotated by genomic coordinates (indicating novel truncations, proteins with retained introns/untranslated regions, previously untranslated regions of the genome, etc.), and 12 protein groups corresponded to known mass spectrometry contaminants. By contrast, the use of the conventional SwissProt FASTA database identified 8004 proteins organized into 4888 protein groups (data not provided).

Differential analysis was performed on the proteogenomics-derived results to associate proteome abundance changes with phenotypic changes in the inflamed tissue samples. A volcano plot of the log_2_ fold-change in protein abundance as a function of -log_10_ corrected *p*-value (Figure 2a) shows that most proteins do not show significant change with *H. hepaticus*-induced colon inflammation.

Differential analysis shows a statistically significant (FDR < 0.05) increase in fourteen murine proteins and a decrease in eight murine proteins (Table 1).

Gene ontology analysis of proteins with an increased abundance in inflamed colon tissue shows enriched GO terms consistent with an inflamed system, showing an enrichment of molecular function GO terms such as MHCI and MHCII complex binding, macrophage migration inhibition factor binding, and oxidoreductase activity, along with the Neutrophil Degranulation reactome and Cd74-Cd44 receptor complex CORUM term (Figure 2b). Proteins that are decreased in abundance in inflamed tissues show enriched GO terms corresponding to molecular functions such as fructose aldolase, the glycolysis/gluconeogenesis and proteosome degradation wikipathway terms, and the 20S proteosome CORUM term (Figure 2c).

Of the proteins found to be significantly increased in abundance in the inflamed proximal colon samples, one protein is unique to the proteogenomic FASTA database. This protein, STRG.18707.1_i_2_260, corresponds to mRNA translated from the (+) strand at chromosome 8, bases 73261429–73261687. This appears to be an untranslated region of the genome which complements the first intron of LARGE Xylosyl- and Glucuronyltransferase 1 (Large 1) (Appendix A). It should be noted that Proteome Discoverer only matched a single peptide QVEIVK at the N-terminus of the purported protein, comprising 7% of the entire protein sequence generated from the RNA-Seq data (Appendix A, Table 1).

### 3.3. Galaxy-P Provides Peptide-Centric Discovery of Non-Canonical Sequences

The isobaric quantitation strategy utilized in the global proteomics strategy is based on abundance measurements of proteins inferred from identified peptides which are labeled with the TMT-reagents; however, a peptide-level analysis is required to further verify and quantify non-canonical peptides belonging to unique proteoforms identified using the proteogenomic database. To this end, an additional workflow was utilized to identify non-canonical peptides in the inflamed proximal colon samples, which could be further verified and validated downstream. Analysis of the protein mass spectrometry data using Galaxy-P using the sectioned proteogenomic FASTA database revealed 14,491 peptides to protein sequences that had no direct sequence match in the canonical SwissProt mouse FASTA database. These peptides were then searched using BLAST-P to detect peptides mapping to the proteins with non-canonical sequences. In filtering these results to remove any matches with 100% alignment to canonical sequences in the reference database, and matches with gaps of zero, the remaining peptide list was reduced to 235 peptides (Figure 3a). These peptides were hypothesized to correspond with novel proteoforms stemming from translation from unexpected genomic locations, splicing events, or non-synonymous coding sequence variants [27].

### 3.4. PepQuery Verifies the Highest Confidence Non-Canonical Peptide Candidates

To verify the variant peptides identified in inflamed proximal colon samples, we used PepQuery v1.3 [40], implemented in Galaxy, on the 235 peptides identified in the discovery workflow. PepQuery provides a rigorous tool to evaluate the confidence of PSMs to non-canonical sequences, via testing for other possible matches (e.g., reference sequences, canonical sequences carrying PTMs) which may better match the MS/MS spectra in question. The list of 235 putative novel, non-canonical peptides was interrogated against the spectra of the TMT-6-labeled fractionated samples and compared to the canonical mouse Uniprot database. Unrestricted modification searching and single amino acid substitutions were performed as a part of the search to detect the strictest matches possible. To be considered passing matches, we used strict criteria where PepQuery had to deliver a *p*-value of <0.05, rank = 1, and the number of unmodified PTM matches set to zero. Of the 235 non-canonical peptides, 58 were found to pass the strict verification criteria (Appendix A, Appendix A) in at least one of the fractionated samples. Of these 58 peptides, only eight were confirmed to be phosphorylated consistent with the original PSM and corresponding to peptides from translated intergenic regions and an assortment of genes (Appendix A). These 58 peptides were largely unique to the Galaxy-P workflow, as none of these peptides was able to be detected in MSFragger with the custom FASTA database and only three peptides—AAAAAAAAAAAAASHSVAK, IQSTNQILEAK, and WTSEFEASLINR—were able to be detected with MaxQuant using the custom FASTA database (Appendix A).

Among the 177 non-canonical peptides that did not pass PepQuery verification, 47 were unmatched by PepQuery to any spectra with sufficient quality scores and were not considered further (Figure 3a). The remaining 130 peptides had either superior matches to peptides in the reference FASTA database, an insufficient *p*-value matching the non-canonical sequence to pass statistical thresholds or matches to reference peptides containing potential PTMs. Interestingly, the non-canonical peptides which did not pass the PepQuery verification are not limited to each of these categories due to the possibility of matching an inputted peptide sequence to an MS/MS spectrum in any of the eight fractionated LC–MS runs in our data. As shown in Figure 3b, most of these non-canonical variants fail verification for multiple reasons, with 34 peptides failing for these three different reasons depending on the LC fraction-specific MS/MS files they were tested against (Figure 3b). Among non-canonical peptides which failed PepQuery verification for a single reason, the majority match to unmodified reference peptides with higher confidence than the non-canonical sequence (Figure 3c), followed by those assigned high PepQuery-derived *p*-values (Figure 3e), with only two peptides being rejected exclusively for matching reference peptides with PTM modifications (Figure 3d).

For the verified non-canonical peptides, the majority were found to be associated with intergenic regions not normally transcribed and translated into proteins (40.85%) as well as introns retained in the translated proteins (28.17%) (Figure 4a). The remaining variant peptides comprise indels, frameshifts, splice junctions, and sequences containing 5′ and 3′ untranslated regions. These peptides are derived from genes and intergenic regions found throughout the genome, excluding chromosomes 6, 18, and 20 (Figure 4b). Gene Ontology analysis of proteins corresponding to those non-canonical sequence peptides found within annotated genes showed no significantly enriched biological pathways common to this set of gene products.

### 3.5. Targeted Proteomics Experiments Validate the Presence of Non-Canonical Peptides

The non-canonical peptides detected using search and verification workflows were found using mass spectrometry data for TMT-labeled, concatenated samples. Because TMT employs protein level-based quantification, we did not have a means to accurately quantify the non-canonical peptide sequences in the control and the inflamed colon samples. We, therefore, ran a separate set of targeted experiments to detect these novel peptide sequences from stored, unlabeled, and unfractionated samples. We used a targeted MS/MS-based parallel reaction monitoring (PRM) assay based on empirically derived *m*/*z* and charge state values from the initial discovery-based analysis. The degree of variant abundance change in the inflamed samples was then expressed as the log2 fold-change of inflamed versus controlled samples, for those peptides displaying confident PRM results (i.e., MS/MS spectra with at least three contiguous product ions in the b- or y-ion series).

Upon re-analyzing the samples, we found that of the 58 non-canonical peptides detected in the original TMT-labeled data, 38 were also detected in the targeted experiments with sufficient confidence (Appendix A). Graphing the log_2_FC of these peptides in inflamed versus control samples shows a general trend of half of the peptides being enriched upon inflammation and the other half being enriched in the control samples (Figure 5a); this pattern was mirrored when comparing the change in peptide abundance with the log_2_FC of the RNA-Seq data of inflamed versus control samples, where there is a very weak correlation between the two (Figure 5b). Ultimately, correcting for multiple hypothesis testing with limma in R found that the changes in abundance of these variants were not statistically significant, though four peptides were found to have uncorrected *p*-values < 0.05 for enrichment or depletion upon inflammation. Of these, three non-canonical peptides showed an increased abundance in inflamed proximal colon samples; these corresponded to an intergenic peptide from chromosome 2 (PIRPGHYPASSPTAVHAIR), a peptide from chromosome 15 stemming from an alternative splicing event (LAHLILSLEAK) and a peptide corresponding to a retained 3-UTR section in Sortilin-related receptor Sorl1 (AASSANIPK, Appendix A). In addition, a non-canonical peptide corresponding to an intergenic region on chromosome 19 was found to be depleted in the inflamed tissue samples relative to the control.

While the differences in abundances of validated non-canonical peptides in inflamed samples and control tissues were not statistically significant, the variant peptides clustered into two groups that show a general trend in increased abundance in the inflamed tissue or increased abundance in the control sample (Figure 5a). There are notable differences between these two groups of peptides. In considering the type of variants present, intergenic regions and introns dominate both groups; however, the variant peptides that show increased abundance in the inflamed tissues are enriched for frameshifts, 3′ UTRs, and indels (Figure 5c). In contrast, the variant peptides found to be decreased in abundance within the inflamed samples (and increased in the controls) contain splice junction variant peptides that are not seen at all in the group showing increased abundance.

## 4. Discussion

In this study, high-resolution mass spectrometry coupled with advanced proteogenomic analysis was utilized to characterize proteome dynamics of proximal colon tissue harvested from mice with chronic inflammation due to infection with *Helicobacter hepaticus.* The results were used to achieve several objectives: (1) Explore the quantitative changes of the proteome upon chronic colon inflammation, including expression levels of non-canonical protein sequences; (2) Develop an integrated bioinformatic and targeted MS-based analytical workflow for verification and validation of non-canonical peptide sequences discovered via proteogenomics; (3) Utilize the knowledge from the verification and validation process as examples of pitfalls related to proteogenomic identification of non-canonical peptides that can inform more accurate studies using this multi-omic approach.

The mouse model utilized in our study, 129S6/SvEvTac-*Rag2^tm1Fw^Il10*^−/−^ (*Rag2*^−/−^*Il10*^−/−^), has been widely used to model inflammatory bowel disease in humans [20,21]. The double knockouts of Recombinase activating gene 2 (*Rag2*) and Interleukin-10 (*Il10*) gene prevent the mice from forming mature T-cells or B-cells or in mitigating the development of chronic inflammation, respectively. As a result, *Rag2*^−/−^*Il10*^−/−^ mice cannot resolve acute inflammation stages and will develop severe chronic inflammation, and eventually cancer, in their colon tissue.

The transition from chronic inflammation to oncogenesis is thought to be one of the subtle changes which occurs through a process of DNA damage accretion [48], epigenetic shifts [49], and eventual phenotypic alteration. This presents a rich landscape for research into biomarkers and therapy for early oncogenesis. In addition, while bottom-up proteomics has found great utility in the study of oncology, the use of conventional genome-derived FASTA databases results in non-canonical protein sequences being missed during data analysis. In this study, we explored the ability of proteogenomics approaches to identify novel protein variants, enabling a more complete characterization of protein dynamics in this model system.

Quantitative proteogenomics analysis utilizing isobaric peptide labeling with the TMT reagent detected several proteins showing increased abundance in the inflamed proximal colon samples. Three of these proteins, haptoglobin, hemopexin, and alpha-1-acid glycoprotein 2, were found to have increased abundance in the serum of *Rag2*^−/−^*Il10*^−/−^ mice with chronic inflammation, being identified in an earlier proteomics study of this model by Knutson et al. [50], indicating their utility as biomarkers for global inflammation; these proteins have also been seen to be increased in abundance in response to sepsis [51], chronic obstructive pulmonary disorder [52], and colorectal cancer [53]. The increased abundance in Prss2, a serine protease involved in the remodeling of the extracellular matrix [54], suggests that the inflamed proximal colon tissue can be considered to be in a chronically inflamed state [55] as increased abundance in Prss2 differentiates IBD patients from healthy patients [56], making the five-month exposure of these mice a suitable model for chronic inflammatory bowel disease. Other indications of chronic inflammation are the increased abundance in the H-2 class II histocompatibility antigen gamma chain Cd74 and the lysosome membrane protein 2 Scarb2, which are indicative of neoantigen generation and presentation to T cells [57]. Other increased proteins consistent with an inflammatory phenotype include heavy-chain Cytochrome b-245 (Cybb), a key component of NADH oxidase in phagocytes needed to create superoxides as a part of the inflammatory response [58], serine palmitoyltransferase 1 (Sptlc1), the initial enzyme involved in sphingolipid synthesis [59] and GTP-binding protein Rheb (Rheb) which serves to activate mTOR1 and promote signal transduction [60]. Interestingly, the increase in abundance of Upp1 seen in the inflamed samples is consistent with the development of many cancers [61,62], indicating a degree of oncogenesis may have begun. These abundance changes to known factors of inflammation demonstrate the accuracy of the TMT-based quantitative proteomics strategy. The loss in abundance of muscle-specific proteins such as Aldoa (fructose-bisphosphate aldolase) and Mustn1 (musculoskeletal embryonic nuclear protein 1) may be due to alteration of the muscularis propria in the proximal colon in response to prolonged inflammation [63].

A major limitation when using TMT-labeling for quantitative proteogenomics is that TMT-based quantitation is protein-centric, inferring protein abundances from peptide sequence matches. When using proteogenomic approaches based on bottom-up MS-based proteomics, matches to non-canonical peptide sequences do not lend themselves to quantitation using this approach. Instead, more peptide-centric analysis is necessary to confirm the presence of these sequences and determine their potential abundance changes, which also reflects differential abundance of the proteoforms to which they belong.

To this end, we employed an advanced peptide-centric proteogenomic bioinformatic workflow to identify non-canonical peptide sequences in an open discovery mode, followed by their verification using the PepQuery tool. The workflow first leverages BLAST-P to see whether putative non-canonical peptide sequences may instead match to other peptides in the conventional proteome; indeed, it was at this step that the STRG.18707.1_i_2_260 peptide QVEIVK was eliminated due to its perfect alignment somewhere else within the mouse proteome. PepQuery enables a rigorous verification of putative non-canonical sequences identified via upstream proteogenomic workflows, addressing a major challenge in proteogenomics to ensure confidence in these identifications [18]. Together, these two nodes of the workflow eliminate false positives of putative non-canonical peptides that are more effectively matched to canonical peptides or common contaminants. There are three ways in which the PepQuery search engine rejects potential non-canonical peptides, all of which were seen in our inflamed proximal colon data and are dependent upon the quality of the PSM within each fractionated mass spectrometry experiment (Figure 3b). In the case of the putative non-canonical peptide AVSPALSIVACSSLAK identified in the first sample fraction, PepQuery can match the spectrum associated with this peptide (Figure 3c, top) as well or better to 44 peptides found within the canonical mouse proteome, including the GTPase Era, mitochondrial isoform peptide SVLLELTAALTEGVVNFK (Figure 3c, bottom), thus rejecting this PSM as identifying a canonical sequence. In another instance from the first fraction, spectra matched to peptides with several repeating residues such as in AGAALPK can potentially have their MS/MS matched to entries in randomized libraries generated in PepQuery, reducing the confidence in the PSM identification (Figure 3e). In this way, PepQuery can eliminate uncertain matches stemming from large mass errors by setting a minimal cutoff value of acceptable match confidences as expressed by *p*-values in the PepQuery outputs. Finally, including additional stringent options in PepQuery, such as unrestricted modification searching and/or amino acid substitution, allows PepQuery to compare “non-canonical” PSMs with reference proteome peptides containing PTMs or amino acid substitutions added in silico, removing the false positive of post-translational modifications to conventional peptides. This option resulted in the rejection of a PSM identifying the non-canonical sequence DIEEIHWFK in favor of a superior match to the canonical MQEQLLEEQK with an a-type ion on the C-terminus corresponding to the loss of part of the C-terminal lysine (Figure 3d). Our results shown in this study provide a cautionary tale to others pursuing bottom-up proteogenomic studies, pointing to the need to carefully verify PSMs to putative non-canonical sequences.

During the final validation via targeted PRM mass spectrometry, 38 of the non-canonical peptides could be detected and quantified by nanoLC-ESI-MS/MS, forming two similarly sized groups of peptides, either showing abundance increase or decrease in the inflamed tissue compared with the controls. These peptides encompass chromosomes throughout the murine genome and represent, principally, the translation of genomic sequences not normally translated, such as intergenic regions, introns, UTRs, etc., indicating potentially altered levels of epigenetic regulation and translational control during colon inflammation [64,65]. Parallel reaction monitoring allowed for the deeper sampling of detected peptides to enable more accurate quantitation as compared with the TMT-based discovery experiments, allowing us to explore the utility of these non-canonical peptides as quantitative indicators of inflammation, or potentially early oncogenesis. Our inability to validate the remaining 21 of our peptide targets could be due to several factors, such as differences between the discovery and validation workflows (different instrument platforms, TMT-labeled peptides detected in the discovery versus non-labeled peptides in the validation, etc.), the lack of suitable peptide standards for targeted method construction or peptide quantitation, potential sample degradation prior to targeted analysis, or interference by co-eluting peaks. These questions make it difficult to determine conclusively whether these sequences were not actually present, or simply were not detectable by PRM. Future studies to answer such questions could include further optimization of a targeted methodology by including synthetic peptide standards, reprocessing of desiccated protein digests that were saved from the initial processing of the inflamed and control proximal colon samples using isotopically labeled internal standards for absolute quantification, in addition to initial optimization of the LC and MS parameters via synthetic peptide standards prior to analysis.

The relevance of these non-canonical peptides detected in mouse proximal colon tissue to human inflammation and oncogenesis was examined via conversion of the mouse genome-coding coordinates for these peptides to analogous human genome coordinates via the LiftOver tool on the UCSC Genome Browser [66]. The human gene sequences were then searched using the online PepQuery server against cancer-tissue derived mass spectrometry data from the Cancer Genome Atlas [30,31,33]. While many non-canonical peptides did not have direct parallels within the human genome or breast, ovarian, and colon cancer datasets from the Cancer Genome Atlas, some sequences queried in the online PepQuery server did show evidence of human variant peptides that were from comparable genetic regions to the variants we observed in our analysis (Appendix A). This demonstrates a potential for these peptides to serve as early biomarkers for human oncogenesis.

Beyond revealing differential protein abundances and sequence variations as a result of colon inflammation and lessons learned in the verification and validation process, a significant deliverable of this work is a novel bioinformatic workflow for discovery and verification of non-canonical peptide sequences identified via proteogenomics. This easy-to-use, open-source and accessible Galaxy-based workflow allows researchers to avoid some of the pitfalls inherent to identifying novel non-canonical peptide sequences. As the workflow is currently focused on verifying novel PSMs, future iterations will incorporate tools for peptide-level quantitative analysis of non-canonical sequences [67].

## 5. Conclusions

In this study, we examined the *Helicobacter hepaticus*-induced inflammation of proximal colon tissue in mice through mass spectrometry-based proteogenomics supplemented with RNA-Seq data. Our initial global proteomics analysis revealed an upregulation of proteins in our inflamed samples consistent with an inflammatory phenotype along with proteoforms that are undetectable using conventional bottom-up proteomics strategies. Through an automated, open access workflow in Galaxy-P, we were able to detect and validate non-canonical peptides across all samples, the majority of which could subsequently be validated using targeted mass spectrometry experiments. We believe this work to be significant in that the workflows presented here allow for the confident identification of non-canonical peptides in mass spectrometry data stemming from insertions and deletions, amino acid substitutions, or alternative splicing events which would serve as invaluable biomarkers for the diagnosis and treatment of colon cancer. The open-source, user-friendly nature of the workflows used in this study allows for their ready uptake and use by non-bioinformaticians, expanding the use of proteogenomics to researchers beyond traditional mass spectrometrists and systems biologists. For future studies, we intend to further optimize the workflows detailed here, allowing for automated quantification of detected non-canonical peptide sequences, as well as automatic generation of parameters for targeted mass spectrometry analysis; in addition, we intend to expand our use of these tools to analyzing other tissues in mice subjected to *Helicobacter hepaticus* infection, such as the distal colon, cecum, and serum samples. In addition, future studies will utilize larger numbers of test animals to increase the statistical power of our analyses. Finally, targeted validation experiments will utilize exclusion lists and extended gradients to detect potential non-canonical peptides more effectively.

## Data Availability

All data were deposited to the ProteomeXchange Consortium via the PRIDE partner repository (http://www.ebi.ac.uk/pride/archive/, accessed on 9 September 2021) and are publicly accessible with the data set identifier PXD028407.

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
