# Peer review of "Quantitative Proteogenomic Characterization of Inflamed Murine Colon Tissue Using an Integrated Discovery, Verification, and Validation Proteogenomic Workflow"

_proteomes, 2022, doi:10.3390/proteomes10020011_

Round 1
Reviewer 1 Report
The work is interesting and well presented. The manuscript should be accepted. Please include the following minor concerns:
- It would be additionally robust to include/verify your findings with another orthogonal standard proteogenomics workflow/method.
- Authors should explain the biological interpretation of novel non-canonical peptides identified from the data.
- Since TMT labeling brings in additional mass tags- assigning PSMs with PTM might be tricky. The PSM figures should contain precursor m/z value, deviation of each ion as well as precursor mass errors in ppm (ideally should be less than 20 ppm)
- The protein abundance changes identified from the TMT data should be reviewed with existing literature and discussed.
Author Response
Reviewer 1
- It would be additionally robust to include/verify your findings with another orthogonal standard proteogenomics workflow/method.
We thank the reviewer for their suggestion and have analyzed our data using MSFragger and MaxQuant to match MS/MS to our RNA-seq generated protein sequencing database. We detected 3 of the 58 peptides in our validated non-canonical peptides were detected in MaxQuant, while none were detected in MSFragger. These results are captured in the new Figure S3 in the Supporting Information and lines 432 through 438 of our manuscript. However, it should be noted that we have verified our peptide results using the PepQuery tool, which validates peptides corresponding to novel proteoforms. The peptide-centric approach used by PepQuery also reduces false positives by consideration of sequence modifications and provides rigorous verification for initial PSM generation. The Pepquery verified peptides were also further validated using targeted PRM analysis, providing analytical confirmation of the presence of these peptides. The inclusion of the MaxQuant and MSFragger results has provided further evidence of our main point in this manuscript - that identification of novel peptide sequences using proteogenomics requires further rigorous downstream steps for verification and validation to ensure confidence in results.
- Authors should explain the biological interpretation of novel non-canonical peptides identified from the data.
In keeping with the reviewer’s comment, we have expanded our discussion on the potential biological relevance of the non-canonical peptides detected in our data in lines 628 through 631 of our manuscript.
- Since TMT labeling brings in additional mass tags, assigning PSMs with PTM might be tricky. The PSM figures should contain precursor m/z value, deviation of each ion as well as precursor mass errors in ppm (ideally should be less than 20 ppm).
We thank the reviewer for their observations and have added the relevant information to Figure S2, as these represent PSMs that were assigned to putative non-canonical peptides in our data.
- The protein abundance changes identified from the TMT data should be reviewed with existing literature and discussed.
We thank the reviewer for their comment; based on this suggestion and those from other reviewers, we have expanded our discussion of the potential significance of those proteins with altered abundances based on knowledge from existing literature within lines 554 through 575 in the discussion section of the manuscript.
Reviewer 2 Report
I think that the herein work provides a great tool especially as it is usable also for the non-bioinformatician.
The authors discuss that 21 peptide target could not be validated and one reason they suggest is "potential issue with sample storage and degradation of the LC column" I appreciate their openess about the subject and understand that it might be hard to re-evaluate especially since experiments take rather long caused by the model system. Since the main aim of the manuscript was the establishment and validation of the pipeline, I think that this is ok for now, but would "produce" more samples in future analyses.
I suggest to shorten the manuscript (especially Discussion + Conclusion), but would include discussion about other "short-term" infections (are proteins/pathways found, different).
Author Response
Reviewer 2
- The authors discuss that 21 peptide targets could not be validated and one reason they suggest is "potential issue with sample storage and degradation of the LC column" I appreciate their openness about the subject and understand that it might be hard to re-evaluate especially since experiments take rather long caused by the model system. Since the main aim of the manuscript was the establishment and validation of the pipeline, I think that this is ok for now, but would "produce" more samples in future analyses.
We thank the reviewer for their insights regarding future analyses. We have updated lines 640 through 648 of the conclusions section of our manuscript to include these recommendations for follow-up studies in this area.
- I suggest shortening the manuscript (especially Discussion + Conclusion) but would include discussion about other "short-term" infections (are proteins/pathways found, different).
We thank the reviewer for their suggestion; in keeping with the recommendations of multiple reviewers, we have shortened sections of the manuscript (removing the final paragraph of the Discussions section due to its redundancy with the Conclusions) and included discussion of changes seen in protein abundance seen in other inflammatory conditions in lines 554-575 of the manuscript.
Reviewer 3 Report
Authors describe a method using an integrated discovery, verification, and validation proteogenomic workflow for the quantitative proteogenomic characterization of inflamed murine colon tissue. The manuscript is well structured and the figures are appropriate for understanding the results. I do have few comments listed below. If the authors address them properly then the manuscript should be accepted for publication.
Check the abbreviations list and make sure you have listed everything including the trivial ones.
In the introduction, try to cite some articles reporting the usage of similar protocols.
The discussion and conclusion contain repetitive statements. Please try to make them concise and avoid repetition.
Author Response
Reviewer 3
- Check the abbreviations list and make sure you have listed everything including the trivial ones.
We thank the reviewer for their comment and have updated the list of abbreviations beginning at line 719 to include the abbreviations we missed in our initial draft.
- In the introduction, try to cite some articles reporting the usage of similar protocols.
We thank the reviewer for their suggestion and have included further citations in the introduction reflecting other sample preparation protocols and proteogenomics applications in lines 68 through 77.
- The discussion and conclusion contain repetitive statements. Please try to make them concise and avoid repetition.
In keeping with the recommendations of multiple reviewers, we have edited the discussion and conclusion sections of the manuscript for content, clarity, and conciseness.
Reviewer 4 Report
Quantitative proteogenomic characterization of inflamed murine colon tissue using an integrated discovery, verification,and validation proteogenomic workflow.
Andrew T. Rajczewski, Qiyuan Han, Subina Mehta, Praveen Kumar, Pratik D. Jagtap, Charles G. Knutson,James G. Fox, Natalia Y. Tretyakova and Timothy J. Griffin.
In this article authors use quantitative proteogenomic to decipher the inflammation process of murine colon tissue and the discovery of new proteins.
The article is well written and workflow well described. This study seems convincing and highlights the difficulty to identify new proteins. This is exemplified with the detection of 235 new protein candidates that has been restrained to 58 after curation to finally identify 39 new proteins. Experimental workflow is in the standard of proteomics studies, as well as the use of standard quantitation protocol. Curation of MSMS data is sounds as exemplified with figure 3C.
However few points must be addressed here.
1- In the material and methods section, additional information on MSMS parameters are needed. This correspond to the mass accuracy, or the exclusion time of an ion for MSMS after fragmentation.
2- Authors mention here the use of inclusion lists to specifically detect and identify new proteins after identification of these last using the dedicated workflow. Why they did not use exclusion lists to repeat the proteomic analyses and go in deeper in the detection of unknown proteins? This a major point knowing that unknown proteins could be drowned in the middle of peptides belonging to canonical proteome. In addition, extending the gradient would help for a better identification of peptides.
3- How calibration was performed ?
4- With how many peptides non-canonical proteins were identified ?
5- Do the proteo-peptides were unique peptides ? This parameter is essential to validate the identification of the non-canonical proteins.
6- Can the authors could explain why there is a 15 min lag between the start and the end of the peak detection / identification in Table S2 ?
7- It is not clear whether the authors identified the peptides using customized protein database and also queried against all major reference databases available for the organism of interest (e.g. RefSeq, UniProtKB, and Ensembl, and also common sample contaminants).
8- How the authors eliminated the most likely sources of false positives (e.g. common posttranslational and chemical modifications, errors in mass measurements, etc.) when reporting novel peptides homologous to a reference sequence ?
9- Authors claimed the use of 3 controls and 3 murine samples. Is that enough to validate the results in terms of biology with the description of up and down regulated biological pathways ?
10- Authors claimed that they searched for post-translational modifications such as phosphorylation. However, no information about the detection of PTM is mentioned in the article.
In light of the article I recommend the publication of this article in Proteomes review after major revisions.
Author Response
Reviewer 4
- In the material and methods section, additional information on MS/MS parameters is needed, including the MS/MS mass accuracy and exclusion time.
We thank the reviewer for their comments. Lines 272 through 276 of the methods section of the manuscript has been updated to contain the QExactive parameters used during the targeted analyses of purported non-canonical peptides, as well as ensuring similar description of parameters used for analysis of TMT-labeled peptides.
- Authors mention here the use of inclusion lists to specifically detect and identify new proteins after identification of these last using the dedicated workflow. Why did they not use exclusion lists to repeat the proteomic analyses and go in deeper in the detection of unknown proteins? This a major point knowing that unknown proteins could be drowned in the middle of peptides belonging to canonical proteome. In addition, extending the gradient would help for a better identification of peptides.
We thank the reviewer for their comments. The proteogenomics experiments presented here were performed as a part of global, untargeted experiments to identify peptides of interest and establish a workflow using both analytical and bioinformatic approaches to verify and validate these peptides. As the reviewer correctly suggests, there are other approaches to probe complex peptide mixtures more deeply and to maximize initial discovery of peptides present in the sample. Although valid, we felt that such experiments were out of scope of the present study.
- How was calibration performed?
We thank the reviewer for their comment. The mass spectrometers used were calibrated using the Pierce Positive Mode Calibration solution. The manuscript has been edited to include this information at lines 93-84, 156-157, and 266-267.
- How many peptides were identified for the non-canonical proteins?
We thank the reviewer for this question. Non-canonical proteins had at least one peptide identified that matched to a novel amino acid sequence within the overall protein sequence. Only one protein containing a novel sequence was significantly increased in the initial global experiment. Ultimately, this peptide did not pass our validation workflow, highlighting the importance of validation steps in proteogenomic analyses.
- Were the non-canonical protein peptides unique to the non-canonical proteins? This parameter is essential to validate the identification of the non-canonical proteins.
We thank the reviewer for their question; non-canonical proteins were considered to be “identified “ if they had at least one peptide that was unique to that sequence. However, this was a different matter to being “validated”, as much of our workflow was devoted to bioinformatically verifying these putative examples. For example, a peptide belonging to our non-canonical protein identified using MSstats ultimately did not pass validation in our validation workflow, indicating a degree of similarity to a peptide belonging to another canonical protein in the mouse proteome. This ultimately resulted in this peptide not being considered further in the paper and not considered a valid biomarker of colon inflammation.
- Can the authors explain why there is a 15 min lag between the start and the end of the peak detection / identification in Table S2?
The information in Table S2 refers to the parameters used for detecting the putative non-canonical peptides using targeted mass spectrometry detection, where the start and end refer to the time windows allowed for detecting the target peptides. An excess of time was used in the detection window to account for any potential drifts in retention time of the non-canonical peptides. In considering the reviewer’s comments, we have edited Table S2 for clarity.
- It is not clear whether the authors identified the peptides using customized protein database and queried against all major reference databases available for the organism of interest (e.g., RefSeq, UniProtKB, and Ensembl, and common sample contaminants).
We thank the reviewer for their comment. In the generation of this custom proteomics database, the resulting sectioned database was concatenated with both the murine UniprotKB FASTA database as well as a selection of common sample contaminants. The manuscript has been updated to clarify these points at lines 221-222.
- How the authors eliminated the most likely sources of false positives (e.g., common posttranslational and chemical modifications, errors in mass measurements, etc.) when reporting novel peptides homologous to a reference sequence?
This is an important observation on the part of the reviewer as the elimination of false positives is of paramount importance in proteogenomics analyses. It is for this reason that the PepQuery search engine was utilized as a part of our workflow to rule out sources of misidentification for our putative non-canonical peptides. By running the search engine in unrestricted modification mode with amino acid substitution enabled, we enabled a possibility of post-translational modifications of murine reference peptides as well as common amino acid substitutions. We have updated our manuscript to clarify the role of PepQuery in our analyses in lines 612-619.
- Authors claimed the use of 3 controls and 3 murine samples. Is that enough to validate the results in terms of biology with the description of up and down regulated biological pathways?
We thank the reviewer for their comments. The use of six samples in this study was in part chosen due to a relatively limited number of samples left from earlier studies performed by our collaborators and to match the samples analyzed in the paper published by Han et al. in the International Journal of Molecular Sciences (https://doi.org/10.3390/ijms22010364) which were used to generate the RNA-Seq-derived FASTA database used in this study. As the purpose of our paper was primarily to demonstrate the utility of our workflow for identifying, verifying, and validating peptides of interest via proteogenomics, we believe that the number of samples was sufficient. The Conclusions section acknowledges a need for further studies to confirm biological relevance of the results presented here, as documented in lines 688 through 691.
- Authors claimed that they searched for post-translational modifications such as phosphorylation. However, no information about the detection of PTM is mentioned in the article.
We thank the reviewer for their comment; the manuscript has been updated to include the inclusion of post-translational modifications, both as an potential modification to the non-canonical sequence as well as through the use of PepQuery for peptide verification when evaluating putatively novel peptide sequences against the murine reference sequence database.
Round 2
Reviewer 4 Report
The present manuscript is accepted in its present form since authors provided enough insights to reviewers requirements.